Transcriptome profiling of anthocyanin-related genes reveals effects of light intensity on anthocyanin biosynthesis in red leaf lettuce

Zhang Yanzhao 1
Xu Shuzhen 1
Cheng Yanwei 1
Peng Zhengfeng 2
Han Jianming 1 hjm@lynu.edu.cn
1 Life Science Department, Luoyang Normal University , Luoyang , China
2 Luoyang Research Institute of Peony , Luoyang , China
Hoecker Ute
Electronic publication date: 2018 Apr 13
Publication date: 2018
Volume: 6
Electronic Location ID: e4607
Received 2017 Nov 12; Accepted 2018 Mar 22
Copyright: © 2018 Zhang et al.
Copyright year: 2018
Copyright holder: Zhang et al.
License: This is an open access article distributed under the terms of the Creative Commons Attribution License, which permits unrestricted use, distribution, reproduction and adaptation in any medium and for any purpose provided that it is properly attributed. For attribution, the original author(s), title, publication source (PeerJ) and either DOI or URL of the article must be cited.
License URL: https://creativecommons.org/licenses/by/4.0/

Keywords: Red leaf lettuce, Transcriptome, Light intensity, Anthocyanins

Funding: National Natural Science Foundation 31400602 Henan Province Science and Technology Breakthrough Project 172102310054 Applied Science and Technology Research Fund of Luoyang Normal University 2015-YYJJ-003 This work was supported by the National Natural Science Foundation (31400602), the Henan Province Science and Technology Breakthrough Project (172102310054), and the Applied Science and Technology Research Fund of Luoyang Normal University (2015-YYJJ-003). The funders had no role in study design, data collection and analysis, decision to publish, or preparation of the manuscript.

==============================
Red leaf lettuce (Lactuca sativa L.) is popular due to its high anthocyanin content, but poor leaf coloring often occurs under low light intensity. In order to reveal the mechanisms of anthocyanins affected by light intensity, we compared the transcriptome of L. sativa L. var. capitata under light intensities of 40 and 100 μmol m−2 s−1. A total of 62,111 unigenes were de novo assembled with an N50 of 1,681 bp, and 48,435 unigenes were functionally annotated in public databases. A total of 3,899 differentially expressed genes (DEGs) were detected, of which 1,377 unigenes were up-regulated and 2,552 unigenes were down-regulated in the high light samples. By Kyoto Encyclopedia of Genes and Genomes enrichment analysis, the DEGs were significantly enriched in 14 pathways. Using gene annotation and phylogenetic analysis, we identified seven anthocyanin structural genes, including CHS, CHI, F3H, F3′H, DFR, ANS, and 3GT, and two anthocyanin transport genes, GST and MATE. In terms of anthocyanin regulatory genes, five MYBs and one bHLH gene were identified. An HY5 gene was discovered, which may respond to light-signaling and regulate anthocyanin structural genes. These genes showed a log2FC of 2.7–9.0 under high irradiance, and were validated using quantitative real-time-PCR. In conclusion, our results indicated transcriptome variance in red leaf lettuce under low and high light intensity, and observed a anthocyanin biosynthesis and regulation pattern. The data should further help to unravel the molecular mechanisms of anthocyanins influenced by light intensity.

Introduction

Color constitutes an important economic trait in vegetables and fruits, and can be mainly attributed to high anthocyanin accumulation. Anthocyanins are the primary pigments in plants and play an important role in the color development, generating a wide range of colors from pink to blue-purple. Anthocyanins have been verified to play a positive role in health, resulting in colorful foods gaining increasing global popularity (Espley et al., 2007; Chiu et al., 2010). The anthocyanin biosynthesis pathway is clear in plants, and three early biosynthesis genes (EBGs), including chalcone synthase (CHS), chalcone isomerase (CHI), and flavonoid 3-hydroxylase (F3H), and four late biosynthesis genes (LBGs) flavonoid 3′-hydroxylase (F3′H), dihydroflavonol reductase (DFR), anthocyanidin synthase (ANS) and 3-glucosyl transferase (3GT) have been identified in the colored tissues of many plants. The enzymes encoded by these genes, in turn, catalyze the substrate to synthesize anthocyanins (Winkel-Shirley, 2001). Anthocyanins are transferred to the vacuoles by glutathione S-transferase and MATE-type proteins (Gomez et al., 2009; Sun, Li & Huang, 2012), where they can function as bioactive molecules and display color. Anthocyanin pathways are mainly regulated by genes from the MYB, bHLH, and WD40 families, and these proteins form a complex by binding to the promoters of structural genes and regulating their transcription (Zhang et al., 2003; Gonzalez et al., 2008; Xu, Dubos & Lepiniec, 2015). The MYB–bHLH–WD40 complex plays a central role in regulating anthocyanins, and many abiotic stresses regulate anthocyanins mainly by activating or inactivating the activity of this complex (Das et al., 2012; Zoratti et al., 2014).

Light is an important factor that influences anthocyanin accumulation in plants, and many experiments have demonstrated that the transcription levels of anthocyanin regulatory and structural genes decrease under light exclusion, which affects the anthocyanin content later on (Zoratti et al., 2014). As observed in apple (Takos et al., 2006; Feng et al., 2013), Chinese bayberry (Niu et al., 2010), and pear (Feng et al., 2010), the bagged fruit display no color, but upon exposure to sunlight, expression of the anthocyanin-related genes are up-regulated, leading to a red pericarp. R2R3 MYBs are important positive regulators that directly influence the expression of the anthocyanin biosynthesis genes. Some R2R3 MYBs have been found to respond to light, such as MdMYB1 in apple (Takos et al., 2006), LrMYB15 in Lilium regale (Yamagishi, 2016), MrMYB1 in Chinese bayberry (Niu et al., 2010), LcMYB1 in litchi (Lai et al., 2014), and VvMYBA1 and VvMYBA2 in grapevine (Azuma et al., 2012). Under changing light conditions, the expression level of these R2R3 MYB transcription factors (TFs) is adjusted to regulate the anthocyanin biosynthesis genes. ELONGATED HYPOCOTYL5 (HY5), a component of light-signaling pathways, has also been linked to the activation of the R2R3 MYBs and key anthocyanin biosynthesis genes in response to light in Arabidopsis and apple (Peng et al., 2013; Shin et al., 2013). Recent research revealed that light signals regulate anthocyanins via CONSTITUTIVE PHOTOMORPHOGENIC1 (COP1), which is a negative regulator and mediate degradation of anthocyanin positive regulators, like PAP1 and PAP2 in Arabiodpsis, MdMYB1 in apples (Li et al., 2012; Maier et al., 2013). Most light signal transduction experiments are analyzed by excluding light. However, Maier & Hoecker (2015) reported that the COP1/SUPPRESSOR OF PHYA (SPA) complex may not be fully inactivated under low light intensity, suggesting the existence of some other light intensity induced mechanism in anthocyanin accumulation.

Leaf color is an important factor influencing the consumer acceptance of red leaf lettuce, and light significantly influences the leaf color (Kang et al., 2013). In our previous work, we discovered that red leaf lettuce displayed a greener leaf color when grown under a light intensity of 40 μmol m−2 s−1, while an increase in the light intensity to 100 μmol m−2 s−1 was associated with a redder color. This red color was caused by increased anthocyanin accumulation. Zhang et al. (2016a) previously reported several putative genes involved in the anthocyanin biosynthetic pathway. However, the molecular mechanisms governing light-induced anthocyanins in red leaf lettuce are still unknown. In this study, we mainly evaluated the effect of light intensity on anthocyanin synthesis using comparative transcriptome analysis. The results should enhance our understanding of the correlation between light intensity and anthocyanin accumulation at the molecular level.

Materials and Methods

Plant materials

Lactuca sativa L. var. “Capitata” was grown in the green house of the Luoyang Normal University (China). Seedlings were grown under white light at an intensity 40 μmol m−2 s−1 and a 16/8 h day/night photoperiod, with the temperature maintained at 22 °C/20 °C. Light intensity was measured using a FGH-1 photosynthetic radiometer (Beijing Normal University Photoelectric Instrument Factory, Beijing, China). Forty days after germination, one group of seedlings was supplied with a light intensity of 100 μmol m−2 s−1, and the other group was used as a control. After three days of treatment, the two leaves at the top of plant were sampled, and the leaves from five individuals were pooled in each group. Samples were frozen in liquid nitrogen and stored at −80 °C. Quantitative (q) analysis of anthocyanin was performed as described by (Zhang et al., 2015), and cyanidin-3-O-glucoside (Cy3G) was used as standards for quantification. The soluble sugar content was detected using the anthrone method (Li, 2003).

cDNA library construction and sequencing

Total RNA was extracted using CTAB-LiCl method (Gambino, Perrone & Gribaudo, 2008), and genomic DNA contamination was removed using DNase I (TaKaRa, Dalian, China). RNA quality was verified by agarose gel electrophoresis and a Bioanalyzer 2100 (Agilent, CA, USA). For cDNA library construction, mRNA was enriched with oligo (dT) magnetic beads and then broken into smaller pieces using fragmentation buffer. The first strand cDNA was reversed-transcribed by random hexamers and small fragment as templates. This was followed by second strand cDNA synthesis using DNA Polymerase I and RNase H. The double strand cDNA was ligated to paired-end adapters, and suitable fragments were selected and enriched with PCR amplification. cDNA libraries were sequenced performed using an Illumina HiSeq2000 platform at BGI Co., Ltd. Sequence data were deposited in the NCBI database with accession numbers SRR5868088 and SRR5943715.

De novo assembly and gene annotation

Raw reads were filtered by discarding the following: reads with adaptors, reads with unknown nucleotides (Nts) larger than 5%, and low quality reads (more than 10% of bases with a Q score ≤ 20). The remaining clean reads from each sample were assembled using Trinity v2.0.6 software (Grabherr et al., 2011). TGICL v2.06 was then used to construct a non-redundant (Nr) unigene set from the two assembled datasets (Pertea et al., 2003). Unigenes were aligned to functional databases to obtain gene functions, mainly including the NCBI Nr, Nt, InterPro, SWISS-PROT, and Kyoto Encyclopedia of Genes and Genomes (KEGG) databases by BLAST v2.2.23 with an E-value threshold <1e−5. Based on the Nr annotation, gene ontology (GO) annotation (http://www.geneontology.org) was performed using the BLAST2GO program v2.5.0 (Conesa et al., 2005).

Analysis of DEGs

Clean reads were mapped to unigenes using Bowtie2 v2.2.5 (Langmead & Salzberg, 2012), and gene expression level was calculated based on fragments per kb per million fragments (FPKM) method in RSEM v1.2.12 (Mortazavi et al., 2008). Significantly differentially expressed genes (DEGs) were scanned among samples under low and high light using EBSeq package v1.7.1 (Leng et al., 2013), with a threshold of an absolute log2 ratio ≥ 2 and a false discovery rate (FDR) significance score <0.001. Based on the KEGG and GO annotation, we classified DEGs and performed functional enrichment using phyper within R (R Development Core Team, 2008). The terms in which FDR not larger than 0.001 are defined as significant enrichment.

Prediction and sequence analysis of TF

ORF of each unigene was predicted using getorf EMBOSS:6.5.7.0 (Rice, Longden & Bleasby, 2000), and hmmsearch v3.0 was adopted to identify TF by aligning ORF to domain from PlantTFDB (Mistry et al., 2013). Neighbor-joining phylogenetic analysis was carried out using MEGA5 (Tamura et al., 2011).

Quantitative-PCR Analysis

Extraction of total RNA and elimination of genomic DNA contamination was performed as above, and 2 μg of total RNA was used as template and the first strand cDNA was synthesized using the PrimeScript™ II first strand cDNA Synthesis Kit (TAKARA, Dalian, China) with primer dT, according to the manufacturer’s instructions. Following 10 times dilution, the cDNA was used for quantitative PCR. qPCR was performed using the SYBR® Premix Ex Taq™ II kit (Tli RNaseH Plus; TAKARA, Dalian, China) according to the manufacturer’s instructions. Gene specific primers are shown in Table S1. The thermal-cycling conditions were as follows: an initial heat denaturing step at 95 °C for 3 min; then 40 cycles of 95 °C for 10 s, 55 °C for 20 s, and 72 °C for 20 s. Each sample was amplified in three independent replicates. Gene transcription levels were calculated using the 2−ΔΔCT comparative threshold cycle (Ct) method (Livak & Schmittgen, 2001), and actin was used as an internal control to normalize the relative expression levels of the analyzed genes (Borowski et al., 2014).

Results

Light intensity determines anthocyanin content

After three days irradiation at a high light intensity of 100 μmol m−2 s−1, the plant turned red, while the plant grown at low light intensity of 40 μmol m−2 s−1 still exhibited a pale green color (Figs. 1A and 1B). High-performance liquid chromatography analysis revealed that anthocyanins accumulated in the plants under high light at a level of 2.1 mg−g, but were barely detected in those grown under low light (Fig. 1C). Our results indicated that anthocyanins were induced as light intensity increased to 100 μmol m−2 s−1, and which resulted in the red coloring in leaves of lettuce. Soluble sugar content was also increased with increasing light intensity (Fig. 1D). The content increased 2.9-fold when irradiated under 100 μmol m−2 s−1 in comparison to 40 μmol m−2 s−1.

Figure 1 Color difference of red leaf lettuce.

(A) Plant was grown under 40 μmol m−2 s−1 light as the control. (B) Plant was irradiated under 100 μmol m−2 s−1 light after three days. (C) Anthocyanins content of red leaf lettuce under different light intensity. (D) Soluble sugar content of red leaf lettuce under different light intensity. Photo credit: Yanzhao Zhang.

Transcriptome sequencing and de novo assembly

After filtering raw data, we generated 59,774,792 clean reads, which included 8,966,218,800 nt from samples under light intensity of 100 μmol m−2 s−1, and 58,856,106 clean reads that included 8,828,415,900 nt from samples under light intensity of 100 μmol m−2 s−1. Trinity software was used to perform de novo assembly with the clean reads, generating 69,733 transcripts with an N50 of 1,500 bp in the high light sample, and 79,037 transcripts in the low light sample with an N50 of 1,460 bp. Redundancy in the transcripts was removed using Tgicl, finally resulting in the generation of 62,111 unigenes with an N50 of 1,681 bp.

A total of 24,129 unigene sequences (38.8%) had a length between 200 and 500 Nts, 13,180 unigenes (21.2%) were between 500 and 1000 nt in length, 16,058 unigenes (25.8%) were between 1,000 and 2,000 nt in length, and 8,744 unigenes (14.1%) were longer than 2,000 nt.

After assembly, gene functions were predicted by querying seven public databases, and a total of 48,435 unigenes (77.98%) were functionally annotated. Among them, 45,046 unigenes (72.52%) obtained hits in the Nr database, 31,786 obtained hits in the SWISS-PROT database, and 33,713 unigenes obtained hits in the InterPro database. TransDecoder v3.0.1 (https://transdecoder.github.io) was used to predict the ORFs. Overall, 22,623 genes were predicted to contain full-length ORFs, accounting for 36.42% of the assembled unigenes. Based on the Nr annotation, the top sequence matches obtained from BLASTX are shown in Fig. S1. The red leaf lettuce sequences had the highest similarity to Cynara cardunculus var. Scolymus sequences (57.02%), followed by Vitis vinifera (5.29%), Sesamum indicum (2.53%), Coffea canephora (1.95%), and Nicotiana sylvestris (1.57%).

Differentially expressed genes

A total of 3,929 unigenes were recovered as differentially expressed between samples grown under low and high light conditions. Compared with the low light sample, 1,377 unigenes were up-regulated and 2,552 unigenes were down-regulated in the high light sample (Fig. 2). To further understand the biological functions of the DEGs, we performed pathway analysis based on the KEGG database. A total of 2,891 unigenes were assigned to six categories including 132 KEGG pathways. KEGG pathway enrichment analysis was performed based on hypergeometric tests. The DEGs between the two samples were significantly enriched in 14 pathways (Fig. 3). Among them, four pathways including “phenylpropanoid biosynthesis (ko00940),” “flavonoid biosynthesis (ko00941),” “flavone and flavonol biosynthesis (ko00944),” and “anthocyanin biosynthesis (ko00942)” are closely associated with anthocyanin biosynthesis, and three glucide metabolic pathways including “other glycan degradation (ko00511),” “starch and sucrose metabolism (ko00500),” and “galactose metabolism (ko00052)” may be related to the synthesis of substrates.

Figure 2 Volcano plot of differently expressed genes between red leaf lettuce under light intensity of 40 and 100 μmol m−2 s−1.

“FDR ≤ 0.01” and “log2 ratio ≥ 1” were used as thresholds to determine the different expressed genes (DEGs). Red points represent up-regulated DEGs, blue points represent down-regulated DEGs, and black points represent non-DEGs.

Figure 3 Pathway functional enrichment of DEGs.

x-axis represents enrichment factor. y-axis represents pathway name. Coloring indicates Q value (high: green, low: red), the lower Q value indicates the more significant enrichment. Point size indicates DEG number (more: big, less: small).

We assigned 1,257 of the 3,929 DEGs to three main GO categories including “molecular functions,” “biological processes,” and “cellular components” (Fig. S2). Among them, 742 unigenes were grouped in the category “cellular components,” 992 unigenes in “molecular function,” and 884 unigenes in “biological processes.” Under the GO category “molecular functions,” the “polynucleotide adenylyltransferase activity” and “oxidoreductase activity” were the most highly enriched terms. Under the category “Molecular function,” the “flavonoid biosynthetic process” were the most highly enriched term.

Analysis of genes involved in anthocyanin biosynthesis and transport

Using gene annotation and phylogenetic analysis, we identified putative structural genes involved in anthocyanin synthesis and transport (Table 1). The accession numbers in GenBank were MF579543–MF579560. In terms of anthocyanin structural genes, a total of nine genes covered each step of the anthocyanin biosynthetic pathway. Among them, the CHS and 3GT gene family contained two members, while the other genes contained only one member. All nine candidate genes were up-regulated under high light, and their normalized transcript levels were two- to ninefold higher than under low light. Among the CHS and 3GT gene family members, Unigene12000_All and CL4808.Contig1_All displayed greater transcript abundance and more obvious growth at the transcription level. It is indicated that Unigene12000_All and CL4808.Contig1_All may be more insensitive to light intensity and contribute more to anthocyanin biosynthesis. In brief, all the putative structural genes were co-up-regulated and the anthocyanin pathway was active under high light conditions.

Table 1 Differentially expressed genes related with anthocyanin.

Gene name	Low_light FPKM	High_light FPKM	Log2FoldChange (high/low light)	FDR	Up/down	
LsCHS	
Unigene12000_All	1.68	858.26	9.0	0	Up	
CL4608.Contig2_All	11.36	97.28	3.1	5.33e−259	Up	
LsCHI	
Unigene10166_All	10.6	68.72	2.70	1.43e−96	Up	
LsF3H	
Unigene8465_All	14.21	408.44	4.85	0	Up	
LsF3′H	
CL524.Contig1_All	9.08	193.8	4.42	0	Up	
LsDFR	
Unigene2105_All	2.3	473	7.68	0	Up	
LsANS	
CL1994.Contig1_All	4.25	269.69	5.99	0	Up	
Ls3GT	
CL4808.Contig1_All	5.84	158.28	4.76	0	Up	
CL4808.Contig2_All	3.31	84.6	4.66	0	Up	
LsGST	
Unigene10814_All	1.84	245.41	7.06	0	Up	
LsMATE	
Unigene12020_All	8.42	18.14	1.11	9.25e−15	Up	
LsMYB	
Unigene12430_All	2.21	8.30	1.91	4.36e−10	Up	
Unigene12294_All	3.14	37.14	3.56	5.02e−85	Up	
Unigene23058_All	0.11	7.25	6.04	5.68e−17	Up	
Unigene24751_All	0.56	22.95	5.36	7.05e−86	Up	
CL6440.Contig1_All	0.72	4.60	2.66	9.14e−11	Up	
LsbHLH	
Unigene13011_All	3.64	21.95	2.59	3.37e−84	Up	
LsHY5	
Unigene19629_All	3.18	10.42	1.71	3.15e−05	Up	

We also detected the differential expression of two anthocyanin transport genes, GST (Unigene10814_All) and MATE (Unigene12020_All), with transcript levels 7.1- and 1.1-fold higher under high light conditions. GST possessed higher transcript abundance and was significantly up-regulated, suggesting that anthocyanins might be primarily transported by GST.

Transcription factors regulating anthocyanin biosynthesis

We predicted a total of 291 TFs belonging to 39 families (Table S2). The MYB gene family represented the largest group containing 33 members, followed by the AP2-EREBP (30 members), MYB-related (26 members), and bHLH gene families (19 members). In model plants, TFs from the MYB, bHLH, and WD40 families regulate transcripts of anthocyanin structural genes. In the 33 MYBs filtered from the detected DEGs, 14 were up-regulated and 19 were down-regulated under high light conditions. Among the up-regulated genes, Unigene12430_All, Unigene12294_All, and Unigene23058_All formed a clade with anthocyanin regulatory genes from Arabidopsis, grape, and Antirrhinum, which have been shown to play a central role in regulating LBGs. Unigene24751_All and CL6440.Contig1_All were closely associated with MYB12 of Arabidopsis, which mainly regulates EBGs (Fig. 4A). Following treatment with high light, the transcription levels of MYBs were up-regulated 1.9- to 6.0-fold.

Figure 4 Phylogenetic analysis of anthocyanin biosynthesis transcription factors of red leaf lettuce.

(A) Phylogenetic tree of five MYBs with MYB transcription factors in other plants. The accession numbers are as follows: AtMYB12 (NP_182268), AtMYB11 (NP_191820), AtMYB111 (NP_199744), AtTT2 (Q9FJA2), AtWER (NP_196979), Ca A (CAE75745), GhMYB10 (CAD87010), LeANT1 (AAQ55181), MdMYB1 (ABK58136), AtMYB0 (NP_189430), MYB75 (NP_176057), MYB90 (NP_176813), MYB113 (NP_176811), MYB114 (NP_176812), NtAN2 (NP_001306786), PhAN2 (AAF66727), PhDPL (ADW94950), PhPHZ (ADW94951), Rosea1 (ABB83826), Rosea2 (ABB83827), VENOSA (ABB83828), VvMYBA1 (AB242302), VvMYBA2 (AB097924). (B) Phylogenetic tree of Unigene13011_All with bHLH proteins in others plants. The accession numbers are as follows: AmDELILA (AAA32663), AtMYC1 (BAA11933), AtEGL1 (NP_176552), AtGL3 (NP_001332705), PhJAF13 (AAC39455), MdbHLH3 (MdbHLH3), MdbHLH33 (DQ266451), PhAN1 (AAG25928), AtTT8 (CAC14865), ZmB (CAA40544), ZmLC (AAA33504), DvIVS (BAM84239), LcbHLH2 (APP94123).

In our study, six bHLH TFs were up-regulated and 13 were down-regulated under high light conditions. Of these, Unigene13011_All was predicted as a anthocyanin regulatory gene and was up-regulated 2.6-fold under high light. Sequence alignment revealed that it contained the BOX18, BOX19, and HLH motifs, which were depicted as conserved in sub group III of the bHLH gene family. It grouped with bHLH in the phylogeny (Fig. 4B), which has been proven to regulated anthocyanin in apple, Petunia, and Arabidopsis, and was most closely related to DvIVS with 62.6% amino acid similarity.

HY5 is a member of the bZIP gene family and acts downstream of the light receptor network and directly affects the transcription of light-induced genes. It was previously verified to regulate anthocyanin structural genes and PAP1 expression by directly binding to their promoters (Shin, Park & Choi, 2007; Shin et al., 2013). In the bZIP family, 14 genes were up-regulated and 19 were down-regulated. Phylogenetic analysis placed Unigene19629_All in the same cluster as the HY5 proteins from different plant species (Fig. S3). Unigene19629_All showed 65.5% amino acid similarity with AtHY5 in Arabidopsis, the transcription level of which was up-regulated 1.7-fold under high irradiance.

Quantitative real-time-PCR validation of DEGs

To further validate the comparative transcriptome results, the transcript level variances of the 18 putative genes involved in anthocyanin synthesis, transport, and regulation between the low and high light conditions were detecting using quantitative real-time (qRT)-PCR analysis. The results indicated that the transcript levels of 17 genes were significantly up-regulated in the high light samples (Fig. 5), which is in agreement with the alterations in gene expression detected by the transcriptome analysis. However, the MYB gene CL6440.Contig1_All was not significantly up-regulated under light intensity of 100 μmol m−2 s−1. Overall, these results indicate that the transcriptomic profiling data correlate with the light intensity responses of red leaf lettuce.

Figure 5 qRT-PCR validation of differentially expressed genes related to anthocyanin.

(A) CHS-1 relative expression level; (B) CHS-2 relative expression level; (C) CHI relative expression level; (D) F3H relative expression level; (E) F3’H relative expression level; (F) DFR relative expression level; (G) ANS relative expression level; (H) 3GT-1 relative expression level; (I) 3GT-2 relative expression level; (J) GST relative expression level; (K) MATE relative expression level; (L) bHLH relative expression level; (M) MYB-1 relative expression level; (N) MYB-2 relative expression level; (O) MYB-3 relative expression level; (P) MYB-4 relative expression level; (Q) MYB-5 relative expression level; (R) HY5 relative expression level. The x-axis represents light intensity (μmol m−2 s−1), and the y-axis represents relative transcription level. All values are normalized relative to the abundance of the actin gene. qRT-PCR analysis were performed in three biological and technical replicates per experiment. The bars represent mean ± SD from triplicate biological repeats.

Discussion

Light is one of the most important environmental factors affecting anthocyanin biosynthesis in plants. Generally, high light intensity is required for the induction of anthocyanin synthesis, and under different light exposure levels, anthocyanin contents have been found to vary in plants and even individual leaves. As found in Lisianthus, the flowers exhibited a paler color under low light conditions, with a 30% reduction in anthocyanin content and 40% reduction in color intensity associated with a 25% decrease from sunlight (Griesbach, 1992). In this study, the leaves of “Capitata” were green under 40 μmol m−2 s−1, but displayed a red pigment when treated with a higher light intensity of 100 μmol m−2 s−1. Our results demonstrated that a certain level of light intensity is necessary for anthocyanin accumulation in red leaf lettuce, which is consistent with previous findings of color variance in lettuce (Voipio & Autio, 1995).

The MYB–bHLH–WD40 complex plays a central role in regulating the anthocyanin pathway, and the environmental factors that affect anthocyanin content usually control the transcript levels of the regulated genes. Many studies have revealed that light mainly affects anthocyanin content via regulation of the transcription levels of MYB-regulated genes, including in apple (Takos et al., 2006), Chinese bayberry (Niu et al., 2010), pear (Feng et al., 2010), and grapevine (Azuma et al., 2012), and the transcription levels of these MYBs change dynamically in response to the light conditions, which is associated with variation in color. In many plants, multiple MYB genes have been shown to redundantly regulate the anthocyanin biosynthesis pathway. For example, in Arabidopsis, PAP1 and PAP2 regulate the late biosynthetic genes, while MYB11, MYB12, and MYB111 mainly regulate the early biosynthetic genes (Stracke et al., 2007). Recently, bHLH genes were reported to be significantly up-regulated when exposed to higher light intensity in Chrysanthemum (Hong et al., 2015), the pericarps of litchi (Zhang et al., 2016b), and peach (Liu et al., 2015). We identified four MYBs and one bHLH gene that were up-regulated under high light conditions, as well as structural genes. Our results indicated that higher light intensity up-regulated the transcript levels of these five genes, and further activated the anthocyanin pathway. MYB1 mainly regulates the late biosynthetic genes, while MYB11, MYB12, and MYB111 mainly regulate the early biosynthetic genes. For the MYB gene CL6440.Contig1_All, a low transcript abundance may explain the variance in transcriptional level obtained by the transcriptome sequencing and qPCR, and it may not be primarily responsible for the regulation of the early anthocyanin pathway.

A ubiquitin E3 ligase CONSTITUTIVE PHOTOMORPHOGENIC1 (COP1) was recently found to repress the activity of positive regulators of anthocyanins at the post-translational level (Li et al., 2012; Maier et al., 2013). In darkness, the COP1/SPA complex localizes to the nucleus where it interacts with the positive regulators of anthocyanins, mediating their ubiquitination and degradation via the 26S proteasome pathway. Conversely, under high light conditions, COP1 dissociates from the COP1/SPA complex via the activated photoreceptors, and is exported from the nucleus. The low COP1 abundance in the nucleus then allows nuclear-localized TFs to accumulate and induce gene expression (Lau & Deng, 2012). Recently research also showed that light affects anthocyanin biosynthesis via transcriptional regulation of COP1. Like in litchi (Zhang et al., 2016b), crabapple (Lu et al., 2016), and eggplant (Jiang et al., 2016), transcript levels of COP1 was decreased from dark to sunlight exposed condition. In our study, the increased light intensity from 40 to 100 μmol m−2 s−1 activated the anthocyanin pathway, but the transcript levels of the COP1 gene were not significantly elevated. We suggested that transcript level of COP1 was not sensitive to variation of light intensity in red leaf lettuce, and the activity of anthocyanin pathway may related with COP1 subcellular localization.

Sugars are endogenous triggers that modulate the expression of anthocyanin biosynthetic genes by acting as signaling molecules and activating anthocyanin regulatory genes by means of a sucrose-specific signaling pathway (Solfanelli et al., 2006). Increasing sucrose concentrations usually induce anthocyanin accumulation, which has even been proposed as a useful phenotypic marker for soluble carbohydrate accumulation (Hu et al., 2002). Several studies suggest that low light intensity affects anthocyanin accumulation through reduced photosynthesis in the leaves or stems, which, in turn, reduces the soluble sugar content of petals and leads to a repression of the genes that encode enzymes of the anthocyanin biosynthetic pathway (Kawabata et al., 1995; Meir et al., 2010). Our KEGG enrichment analysis revealed four sugar pathways and four anthocyanin-related pathways, which indicated that a higher light intensity of 100 μmol m−2 s−1 increased soluble sugar concentration during photosynthesis, which not only provides a substrate for anthocyanins, but also activates signals to regulate the expression of anthocyanin-related genes.

Conclusion

In red leaf lettuce, anthocyanin content is regulated by light intensity. Under low light conditions (40 μmol m−2 s−1), the anthocyanin pathway is inactive. When the light intensity is increased to 100 μmol m−2 s−1, the putative genes corresponding to anthocyanin biosynthesis, transport, and regulation were significantly up-regulated. Glucide metabolic may play important role in anthocyanin accumulation as the increased light intensity.

Supplemental Information

Supplemental Information 1 Species distribution of the BLASTX results.

This figure shows the species distribution of unigene BLASTX results against the nr protein database with a cutoff E value <10−5 and the proportions of each species. Different colors represent different species.

Click here for additional data file.

Supplemental Information 2 GO categories of the unigenes.

Unigenes were annotated in three categories: cellular components, molecular functions, biological processes.

Click here for additional data file.

Supplemental Information 3 Phylogenetic tree of red leaf lettuce HY5 with bZIP members from other plants.

Arabidopsis bZIPs were down load from PlantTFDB database (http://planttfdb.cbi.pku.edu.cn/).

Click here for additional data file.

Supplemental Information 4 Primers used for qRT-PCR.

Click here for additional data file.

Supplemental Information 5 Transcription factors in DGEs.

Click here for additional data file.

We thank Huiping Ma of Luoyang Research Institute of Peony (Luoyang, China) for providing advice for experiments.

Additional Information and Declarations

Competing Interests

Author Contributions

DNA Deposition

Data Availability

The authors declare that they have no competing interests.

Yanzhao Zhang conceived and designed the experiments, performed the experiments, analyzed the data.

Shuzhen Xu performed the experiments, contributed reagents/materials/analysis tools.

Yanwei Cheng performed the experiments, prepared figures and/or tables.

Zhengfeng Peng analyzed the data, contributed reagents/materials/analysis tools, prepared figures and/or tables.

Jianming Han conceived and designed the experiments, authored or reviewed drafts of the paper, approved the final draft.

The following information was supplied regarding the deposition of DNA sequences:

Anthocyanins related genes described here are accessible via GenBank accession numbers MF579543 to MF579560.

The following information was supplied regarding data availability:

Raw data were deposited in the NCBI database with accession numbers SRR5868088 and SRR5943715.

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
