# Peer review of "Transcriptome profiling of anthocyanin-related genes reveals effects of light intensity on anthocyanin biosynthesis in red leaf lettuce"

_PeerJ, doi:10.7717/peerj.4607_

## Round 0.1 · original submission · Major Revisions

· Academic Editor

Major Revisions

Please address all reviewers´ comments, paying particular attention to their comments on the quality control of the transcriptome data, the disussion section of your manuscript and the quality of the English.

·

Basic reporting

The manuscript’s structure conforms to PeerJ’s standards and the figures are relevant and of sufficient quality. However, there are a number of issues regarding the reporting that need to be addressed:

1. The NCBI SRA datasets SRR5868088 and SRR5943715 are not accessible. Trying to access them results in an error message stating that “This record has not yet been released. If you found reference to this accession in a publication, please let us know”. This is not in accordance with journal policy and the raw data should be made accessible for review.

2. The sentence structure and phrasing requires considerable editing. Some instances where the language/phrasing is imprecise or ambiguous are listed below:
2a) When referring to the identified lettuce gene homologues, it should be made clear that these are putative anthocyanin structural or regulatory genes as their involvement in the pathway is only inferred from phylogeny and correlation of gene expression. This is done in most cases, but not everywhere (e.g. lines 174, 183, 214, 281)
2b) It is not always clear whether a statement refers to anthocyanin biosynthesis, accumulation or the expression of anthocyanin-related genes (e.g. lines 64, 76)
2c) On several occasions the text reads “anthocyanin-regulated” genes (i.e. regulated by anthocyanins) when it should refer to genes regulating anthocyanin levels (lines 38, 188, 195, 201, 268).
2d) The sentence referring to Unigene12000_All and Unigene10814_All in lines 180-182 is not comprehensible.

3. The introduction could benefit from adding a few additional pieces of information which will make the article more accessible to non-experts
3a) A sentence or two stating what anthocyanins are and from what they are synthesized might be helpful. The abbreviations EBGs and LBGs should be introduced here.
3b) Two putative homologues of anthocyanin transporters are mentioned in the results and in the abstract. Thus, anthocyanin transport should be mentioned in the introduction and references for this should be given.
3c) References should be given for the statement in lines 59-61. Given their importance in the manuscript, it could also be established here that the MYB transcription factors and HY5 control light-dependent expression of the structural genes in other species.

4. There appears to be some confusion regarding the term “-fold” or “fold change”. While fold change (FC) in transcript levels, i.e. a ratio of gene expression values, is often stated for qPCR results, log2FC is commonly used to refer to gene expression changes in RNA-seq analysis. It should be clearly stated whether the numbers mentioned in the text represent FC or log2FC so that readers do not underestimate the observed differences.

5. Gene names should be stated in full the first time they occur.

Experimental design

The presented research is within the scope of the journal. The experimental design of this study is sound, the applied methodology and bioinformatics analyses are routinely used. Below are some suggestions that may further improve the quality of the manuscript.

1. Has the sequencing data been examined for contamination? As the lettuce is grown in non-sterile conditions, bacterial, fungal etc. transcripts can represent a significant amount of the detected sequences. Checking the species distribution among the BLASTX hits could reveal the extent of potential contamination. If there is a large amount of reads that represent hits for non-plant sequences, filtering the data against the respective genomes should be considered.

2. The authors might consider to annotate the differentially expressed genes not only for KEGG pathways, but also for GO terms or COG categories, which are commonly used alongside KEGG in the literature. Alternatively, they should state why KEGG is the database of choice for their analysis.

3. The software packages used for differential gene expression analysis need to be stated. Version numbers should be stated for all mentioned software packages.

4. Quantification of anthocyanin levels and basic statistics on this could be included in Figure 1.

5. A reference for the CTAB method (line 92) should be given.

Validity of the findings

The results obtained in this study are clearly related to the previously defined research question and significantly advance the understanding of the regulation of anthocyanin accumulation in red leaf lettuce. Some conclusions however need clarification:

1. A putative lettuce homologue of Arabidopsis HY5 has been identified – is there phylogenetic information supporting this? Is HY5 the best hit when BLASTed against Arabidopsis proteins and is Unigene19629_All the best hit when the AtHY5 sequence is BLASTed against the assembled unigenes?

2. Why do the authors contrast the lack of change in lettuce COP1 transcript levels in lines 256-259 with the regulation of COP1 in Arabidopsis? Light does not strongly affect Arabidopsis COP1 transcript, but affects the protein’s interactions, activity and subcellular localisation. Hence, a major effect of light on lettuce COP1 transcript would not be expected.

3. The paragraph in lines 260-266 is not clearly laid out and the hypotheses stated in it are vague – what exactly are the PSI and PSII pathways? It could be left out of the discussion or – if the authors feel it represents an important conclusion drawn from their data – should be more carefully explained with precise hypotheses and arguments.

Additional comments

In their manuscript, Zhang et al. compare transcriptomes of low and high light-grown red leaf lettuce to detect transcripts correlated with increased anthocyanin content under high light; using gene annotation and phylogenetic analysis they identify genes encoding putative anthocyanin biosynthesis enzymes, anthocyanin transporters and transcription factors regulating anthocyanin accumulation.

Overall, the manuscript presents valuable information on light-regulated anthocyanin biosynthesis in red leaf lettuce that may also serve as basis for further investigations. There is a number of issues with regard to the reporting of the findings that require editing; however, if these issues are appropriately addressed and if the data sets are made accessible, the manuscript should be considered for publication.

Reviewer 2 ·

Basic reporting

The manuscript, "Transcriptome profiling of anthocyanin-related genes reveals effects of light intensity on anthocyanin biosynthesis in red leaf lettuce" attempts to investigate the genetic basis of light-mediated anthocyanin biosynthesis in red leaf lettuce using transcriptomic approach.

The authors first establish the effect of increased light intensity on the coloration of red leaf lettuce, followed by transcriptomic analysis to identify genes associated with increased anthocyanin accumulation under higher light intensity.

The manuscript needs improvement at a few places for general writing and meaning.
- Line 71-75: Rephrase the sentence for better readability.
- Line 280-282: Sentence may be broken down to two sentences.
- Very long sentences with many commas should be avoided.
- Line 52: "The enzymes encoded by these gene..."
- Line 38 and 60: "anthocyanin-regulatory"

Experimental design

- Line 88 mentions that, "leaves from five individuals were pooled in each group...". But there is no mention of number of biological replicates used for the study. There must be at least three biological replicates for the statistical analysis, and FDR cutoff used in the study. Authors must include a scatter-plot showing the distribution if replicates for each sample.
- What is the reasoning for doing two independent assemblies for low light and high light conditions? Since the organism and tissue type are same, it would have been better to do a single assembly combining all the reads to increase the chances of getting full length ORFs.
- Did authors do something to check the quality of assembled transcriptome? It is currently not mentioned in the manuscript. Perhaps estimating the percentage of full length ORFs as explained in Trinity manual might be performed.
- Line 107-109: Blastx can not be performed against nucleotide (Nt) database. Authors should make sure about the correctness of the statement. I feel Blastx might be performed against Nr, SWISS-PROT and InterPro databases as also mentioned in line 156-158.
- Figure 3: Legend mentions white and blue colors, but figure has green and red.

Validity of the findings

From the transcriptomic analyses, authors find some relevant anthocyanin biosynthetic and transport genes upregulated in response to increased light intensity. In addition, authors also find the differential expression of some key transcription factors regulating the anthocyanin biosynthesis. The authors validate some of those results using qRT-PCR. However there are many sections in the discussion that seem to be over-statement.
- There is no results presented about the photosystem (PS) I and PS II. Hence, the discussion paragraph (line 260-266) seems to be out of place.
- The discussion about COP1/SPA complex also doesn't well, as the paper talks about the transcript levels. But COP1/SPA mediated regulation happens at the level of protein. Therefore, the section should be rephrased to make it more contextual.
- There is not enough evidence for the discussion regarding the light-mediated increase in sugar levels, which in turn increased anthocyanin levels. Either quantification of sugar levels under different light conditions or expression analysis of critical sugar-related genes need to be performed to justify the statements.

---

## Round 0.2 · Minor Revisions

· Academic Editor

Minor Revisions

Please address the remaining reviewers´ comments in a revised version.

·

Basic reporting

The authors have made the changes requested and the language of the manuscript has improved considerably. There are two minor issues that need to be addressed:

1. The authors have clarified when referring to fold change (FC) or log2FC. However, I think the wording for log2FC needs to be slightly altered; levels can be “x-fold higher/upregulated”, but not “x-log2 fold change higher/upregulated” (lines 40, 209, 216, 229, 233, 244). As a log2FC > 0 already implies increase, the text could simply read “genes showed a log2FC of 2.7-9.0” (example for line 40).

2. It appears that two different unigenes are referred to as Unigene10814_All (Table 1 and lines 210 and 216), one a 3GT and the other a GST homologue. This needs to be clarified.

Experimental design

No additional comments

Validity of the findings

No additional comments

Additional comments

The authors have satisfactorily addressed my previously raised concerns and made the recommended changes.

Reviewer 2 ·

Basic reporting

Authors have addressed the comments properly, and the revised manuscript is significantly improved over the previous version.

Experimental design

Authors have sufficiently addressed my comments for previous version of the manuscript, except for the usage of the replicates in the experimental design. If the authors have taken one pool of each sample for RNAseq library preparation and analysis, I am not sure about the usage of FDR cutoff for selecting the genes. Without having three replicates for the analysis, the analysis is just qualitative and FDR cutoff doesn't have any meaning. Since the authors have checked the candidate genes by qPCR, the interpretation in the paper OK. However, authors should be careful using FDR cutoff in text.

Validity of the findings

Authors have done a few new experiments ( e.g. sugar content measurement) and analyses (e.g. prediction of full lenght ORFs) to justify the validity of findings in the revised manuscript. Further, authors have removed some of vague and speculative statements from the discussion regarding the findings.

---

## Round 0.3 · accepted · Accept

· Academic Editor

Accept

Thank you for addressing the reviewers´ comments and for making the appropriate changes in the manuscript.

#